# Development of type 2 diabetes in women with comorbid gestational diabetes and common mental disorders in the Born in Bradford cohort

Claire A Wilson ![ORCID] ,[1,2] Gillian Santorelli,[3] Rebecca M Reynolds,[4] Emily Simonoff,[2,5] Louise M Howard,[1,2] Khalida Ismail[2,6]

LMH and KI are joint senior authors.

For numbered affiliations see end of article.

**Correspondence to**
Dr Claire A Wilson;
claire.1.wilson@kcl.ac.uk

## ABSTRACT

**Objectives** To compare, in a population of women with gestational diabetes mellitus (GDM), the time to diagnosis of Type 2 diabetes in those with and without common mental disorder (CMD) (depression and/or anxiety) during pregnancy.

**Design and setting** prospective study of the Born in Bradford cohort in Bradford, UK.

**Participants** 909 women diagnosed with GDM between 2007 and 2010, with linkage to their primary care records until 2017. The exposed population were women with an indicator of CMD during pregnancy in primary care records. The unexposed were those without an indicator.

**Outcome measures** Time to diagnosis of type 2 diabetes as indicated by a diagnosis in primary care records.

**Analysis** time to event analysis using Cox regression was employed. Multiple imputation by chained equations was implemented to handle missing data. Models were adjusted for maternal age, ethnicity, education, preconception CMD and tobacco smoking during pregnancy.

**Results** 165 women (18%) were diagnosed with type 2 diabetes over a follow-up period of around 10 years. There was no evidence of an effect of antenatal CMD on the development of type 2 diabetes following GDM (adjusted HR 0.95; 95% CI 0.57 to 1.57).

**Conclusions** Women with CMD were not at an increased risk of type 2 diabetes following GDM. This is reassuring for women with these co-morbidities but requires replication in other study populations.

## INTRODUCTION

Gestational diabetes mellitus (GDM) is defined as diabetes which occurs for the first time during pregnancy and affects between 5% and 10% of pregnant women.[1 2] The common mental disorders (CMD) of anxiety and depression are also highly prevalent during pregnancy, affecting at least 10% of women.[3 4] There is now evidence to support an association between GDM and antenatal (during pregnancy) depression,[5] meaning that the two conditions may be frequently comorbid. Both conditions are associated

### Strengths and limitations of this study

► First study to investigate the impact of mental disorder during pregnancy in women with gestational diabetes on subsequent type 2 diabetes risk.
► Large cohort study of a multiethnic UK population of 909 women with gestational diabetes.
► Comprehensive follow-up over a decade via linkage to primary care records.
► The moderating effect of insulin use and levels of hyperglycaemia at diagnosis on the outcome of time to development of type 2 diabetes was also assessed.
► Unable to study the influence of other factors related to mental health such as medication adherence and lifestyle management on the risk for type 2 diabetes.

with adverse obstetric, neonatal and longer-term outcomes for women and their children.[4 6]

Up to 60% of women with GDM will develop type 2 diabetes at some point in their lifetime following pregnancy.[7] This is thought to be due to the pathological process of insulin resistance common to both GDM and type 2 diabetes. Pregnancy itself is a state of insulin resistance. While this may resolve following pregnancy, there is a risk of advancing insulin resistance in some women who have experienced GDM as they age. Other risk factors for development of type 2 diabetes include socioeconomic deprivation and non-white ethnicity.[8]

There are also clinical markers of risk during pregnancy for the future development of type 2 diabetes. These include fasting glucose at the time of the OGTT (oral glucose tolerance test): the diagnostic test for GDM.[8] Yet there is evidence that glycaemic control during pregnancy in those with GDM impacts future type 2 diabetes risk independently of baseline fasting glucose at the time of GDM

diagnosis.[9 10] There is also evidence of an increased risk in women whose GDM has required the use of insulin; this may be a marker of increased severity of insulin resistance, relative pancreatic insufficiency and/or poorer glycaemic control post diagnosis.[11 12] In the non-pregnant population of those with type 2 diabetes, hyperglycaemia was significantly associated with depression in a meta-analysis of 24 studies.[13] Indeed there is evidence for a relationship between depression and various factors known to impact glycaemic control; those with depression and type 2 diabetes may be less likely to adhere to lifestyle interventions, such as those involving diet and physical activity[14] and also medication[15]: both of which are also elements of GDM management.[16] Thus, it may be hypothesised that in women with GDM, comorbid CMD during pregnancy could increase the risk of type 2 diabetes in the future due to poorer glycaemic control during pregnancy.

Despite this, none of the recent risk prediction models for the development of type 2 diabetes following GDM have included mental disorder.[17] Greater understanding of the impact of antenatal CMD on risk for subsequent type 2 diabetes in women with GDM could provide further insights into the risk profile of women who are more likely to develop type 2 diabetes. Knowledge of such modifiable risk factors could inform interventions to reduce this risk. The aim of this study was to compare, in a population of women with GDM, the time to development of type 2 diabetes in those experiencing antenatal CMD, with those not experiencing antenatal CMD.

## METHODS
### Sample: women with GDM)
Born in Bradford (BiB) is a prospective longitudinal cohort of 12 450 women with 13 758 pregnancies in Bradford. Bradford is a city in the north of England and has high levels of deprivation; 60% of babies BiB are born into the poorest 20% of the English and Welsh population according to Index of Multiple Deprivation (IMD 2010). Its ethnic makeup is predominantly biethnic: Pakistani and white British.

Pregnant women were recruited to the BiB cohort between 2007 and 2010,[18] when attending a routine appointment for a 2-hour 75 g OGTT, as currently recommended by the UK's National Institute for Health and Care Excellence as the gold standard for diagnosing GDM.[16] It is offered to all women in Bradford, usually between 26 and 28 weeks gestation. Consent was obtained for record linkage to maternity and primary care records via SystmOne: a clinical computer system used by almost all general practices in Bradford and which provides primary care data on diagnoses and prescriptions.

GDM was diagnosed according to modified WHO criteria (either fasting glucose ≥6·1 mmol/L or 2-hour postload glucose ≥7·8 mmol/L).[19] The sample used for analysis was women with GDM and linkage to primary care records (n=994; see figure 1). If women had more than one GDM affected pregnancy in the cohort, the

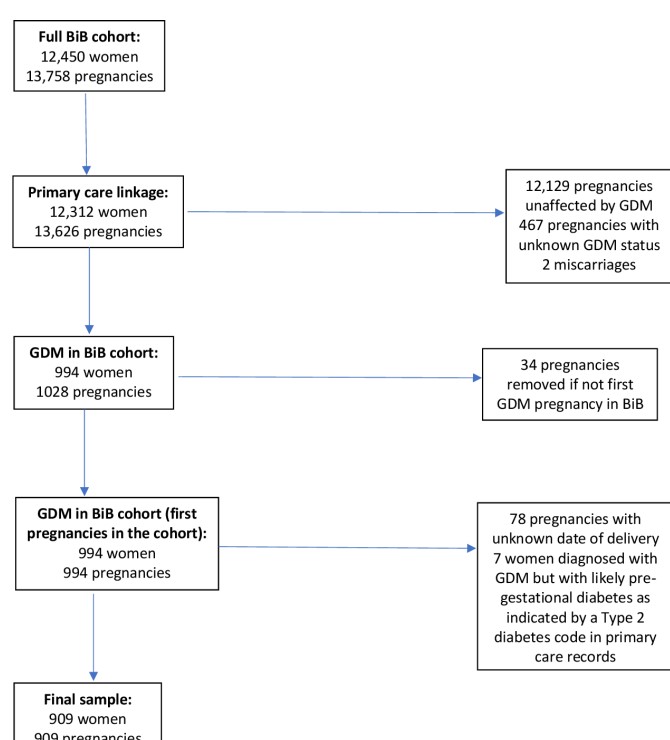

**Figure 1** How the sample was obtained. BiB, Born in Bradford; GMD, gestational diabetes mellitus.

first pregnancy (and its date of delivery) was used as the start of the 'at risk' period for type 2 diabetes. Women with unknown date of delivery also were excluded due to being unable to determine the start of their 'at risk' period. This gave a final sample of 909 women.

### Exposure: antenatal CMD
As an indicator of CMD, Read codes (Clinical Terms V.3, CTV3) for diagnosis and/or treatment of CMD were extracted from general practitioner (GP) electronic health records (EHR) alongside medication prescriptions as per previously published methods.[20] These relate to International Classification of Diseases (ICD-10) diagnostic groups F32 and F33 depressive episodes and recurrent depressive disorders and F41 anxiety disorders.[21] The binary variable of antenatal CMD was defined as any indicator of CMD appearing in the primary care notes between the estimated date of conception (estimated from date of last menstrual period and/or booking scan) and the date of delivery. Further information on the codes used is provided in online supplemental material S1.

### Outcome: time to development of type 2 diabetes
Read code (CTV3) for 'type 2 diabetes mellitus': X40J5 was used as an indicator of type 2 diabetes development.[22] The outcome was time to the earliest recorded code in the primary care records after the date of delivery of the GDM affected pregnancy (in years).

### Covariates
There were a number of variables identified as potential confounders due to their association with both GDM and

CMD but not on the causal pathway from exposure to outcome.[23] These were maternal age[8 24] at the time of the OGTT and tobacco smoking during pregnancy.[25 26] CMD occurring prior to pregnancy ('preconception' CMD) was also measured using Read codes and medication prescriptions in the primary care record from the woman's birth to the estimated date of conception of the woman's first ever pregnancy (see online supplemental material S1). Also included were the sociodemographic characteristics of maternal ethnicity (as a three category variable of Pakistani, White British or other minority ethnic group) and socioeconomic status (SES)[8 26 27]; rather than using IMD[28] to assign SES, a five category variable of maternal education was used a proxy, as the high levels of deprivation in BiB result in a highly skewed distribution of participants across deprivation categories. It was also hypothesised that the continuous variable of fasting glucose on the OGTT and the binary variable of GDM treated with or without insulin may be effect modifiers.

## Statistical analysis

Data were analysed using Stata V.15.[29] Survival analysis using Cox regression was used to compare time to development of type 2 diabetes between exposed and unexposed groups, adjusting for relevant confounders, to produce a HR.[30] The time window was date of delivery of GDM pregnancy to either date of type 2 diabetes development or date of the last EHR extraction (9 August 2017). Interaction terms for the hypothesised effect modifiers were also included in the models. Kaplan-Meier failure curves were produced showing time to development of type 2 diabetes between women with and without antenatal CMD.

Multiple imputation by chained equations was implemented to handle missing data.[31] 20 imputations were used according to the proportion of participants with any missing data.[32] All analysis variables, including the time variable, were included in the imputation model.[33] Estimates were obtained by pooling results using Rubin's rules.[34] Complete case analyses were also conducted.

A sensitivity analysis removing women with co-morbid severe mental illnesses (SMI) was also conducted. These relate to ICD-10 diagnostic groups F20-29 psychotic disorders including schizophrenia, F30-31 bipolar disorder and F60 personality disorders[21] using GP EHR alongside medication prescriptions as per previously published methods.[20] Further information on the codes used is provided in online supplemental material S2.

## Patient and public involvement

Women with lived experience of GDM and/or mental disorder during pregnancy were consulted from study inception to completion of the analysis. They indicated that the research question was important to them. They also informed interpretation of the study's results; for example by indicating that they found the findings reassuring. These women came from the Service User Advisory Group of the Section of Women's Mental Health at

King's College London and the BiB Parent Governors Group. The authors thank them for their contribution.

## RESULTS

Characteristics of women included in this analysis, stratified by development of type 2 diabetes, are presented in table 1. Fifty-nine per cent of women with data on ethnicity were of Pakistani ethnic origin and mean maternal age was 30 years (SD 5.5). Twenty-eight per cent of women with data on education had higher than A level (secondary school) qualifications. Nine per cent of women self-reported tobacco smoking at some point during their pregnancy. Eleven per cent had an indicator of antenatal CMD. Fifteen per cent of those with available data had an indicator of preconception CMD. Eighteen per cent of women with GDM developed type 2 diabetes during the follow-up period. Pakistani ethnicity, older age and lower education were associated with the development of type 2 diabetes.

The median time taken for 15% of women to develop type 2 diabetes in the group unexposed to antenatal CMD was 6.5 years (95% CI 4.9 to 7.9) vs 7.4 years (95% CI 3.6 to 7.9) in the exposed group. Table 2 displays the unadjusted and adjusted results of the Cox regression for type 2 diabetes in women with and without an indicator of antenatal CMD, using imputed data. There was no evidence of an effect of antenatal CMD on the development of type 2 diabetes following GDM on unadjusted or adjusted analyses. These findings are illustrated in figure 2, which shows Kaplan-Meier failure curves for time to development of type 2 diabetes between women with and without antenatal CMD. Results of complete case analyses mirrored those of the imputed results (see online supplemental material S3). There was no evidence for an interaction between antenatal CMD and fasting glucose (p=0.555) or between antenatal CMD and insulin-treated GDM (p=0.680). Removal from the sample of 57 women with comorbid SMI resulted in little change to the effect estimate (unadjusted HR 0.95 (95% CI 0.55 to 1.65) and adjusted HR 0.89 (95% CI 0.51 to 1.56).

## DISCUSSION

There was no evidence in this cohort of women with GDM that antenatal CMD increased the risk of subsequent type 2 diabetes. The proportion of women in this sample developing type 2 diabetes was similar to that observed in a number of previous studies, which report between 10% and 20% of women with GDM developing type 2 diabetes over similar follow-up periods of around 10 years.[35–37] There was also no evidence that fasting glucose at the time of GDM diagnosis or insulin use, as indicators of the severity of insulin resistance, modified associations between antenatal CMD and type 2 diabetes.

This is the first study to investigate the impact of mental disorder during pregnancy in women with GDM on subsequent type 2 diabetes risk. It is also one of the

**Table 1** Characteristics of the sample (N=909 women)

| | Total | | Type 2 diabetes | | No type 2 diabetes | | P value* |
|---|---|---|---|---|---|---|---|
| | N=909 | % | N=165 (18.2%) | % | N=744 (81.8%) | % | |
| **Ethnicity** | | | | | | | |
| Pakistani | 472 | 51.9 | 102 | 61.8 | 370 | 49.7 | <0.001 |
| White British | 198 | 21.8 | 15 | 9.1 | 183 | 24.6 | |
| Other minority ethnic group | 128 | 14.1 | 24 | 14.6 | 104 | 14 | |
| Missing | 111 | 12.2 | 24 | 14.6 | 87 | 11.7 | |
| **Maternal age (years)** | | | | | | | |
| Mean (SD) | 30.3 (5.5) | | 32.2 (5.3) | | 29.9 (5.5) | | <0.001 |
| Missing | 41 | 4.5 | 7 | 4.2 | 34 | 4.6 | |
| **Maternal education** | | | | | | | |
| Less than 5 General Certificate of Secondary Education (GCSE) equivalents | 206 | 22.7 | 55 | 33.3 | 151 | 20.3 | <0.001 |
| 5 GCSE equivalents | 218 | 24.0 | 37 | 22.4 | 181 | 24.3 | |
| A level equivalents | 87 | 9.6 | 12 | 7.3 | 75 | 10.1 | |
| Higher than A level | 223 | 24.5 | 24 | 14.6 | 199 | 26.8 | |
| Other | 63 | 6.9 | 13 | 7.9 | 50 | 6.7 | |
| Missing | 112 | 12.3 | 24 | 14.6 | 88 | 11.8 | |
| **Maternal tobacco smoking in pregnancy** | | | | | | | |
| Yes | 73 | 8.0 | 15 | 9.1 | 58 | 7.8 | 0.470 |
| No | 722 | 79.4 | 124 | 75.2 | 598 | 80.4 | |
| Missing | 114 | 12.5 | 26 | 15.8 | 88 | 11.8 | |
| **Preconception CMD** | | | | | | | |
| Yes | 123 | 13.5 | 14 | 8.5 | 109 | 14.7 | 0.056 |
| No | 710 | 78.1 | 131 | 79.4 | 579 | 77.8 | |
| Missing | 76 | 8.4 | 20 | 12.1 | 56 | 7.5 | |
| **Antenatal CMD** | | | | | | | |
| Yes | 95 | 10.5 | 17 | 10.3 | 78 | 10.5 | 0.945 |
| No | 814 | 89.6 | 148 | 89.7 | 666 | 89.5 | |
| **Fasting glucose (mmol/L)** | | | | | | | |
| Mean (SD) | 5.2 (1.09) | | 6.2 (1.5) | | 5 (0.83) | | <0.001 |
| Missing | 41 | 4.5 | 7 | 4.2 | 34 | 4.6 | |
| **Insulin treated GDM** | | | | | | | |
| Yes | 420 | 46.2 | 47 | 28.5 | 373 | 50.1 | <0.001 |
| No | 489 | 53.8 | 118 | 71.5 | 371 | 49.9 | |

*On $\chi^2$ for categorical variables and t-test for continuous variables comparing groups with and without type 2 diabetes for complete cases with no missing data on exposure.
CMD, common mental disorders; GDM, gestational diabetes mellitus.

largest samples in the UK reporting type 2 diabetes risk following GDM.[12] The sample was drawn from an urban, deprived and multi-ethnic population. While older women and women of Pakistani ethnicity were relatively over-represented in this sample compared with both the BiB cohort as a whole and the city of Bradford,[18] this is representative of those women at risk of GDM so is not unexpected.[6 16] These same demographic predictors of type 2 diabetes and the proportions of the sample with CMD and type 2 diabetes indicate face validity of the measures of both exposure and outcome.

Nonetheless, measures taken from primary care records are reliant on individuals attending primary care services. For example, a number of studies have documented the underdiagnosis of CMD in primary care,[38 39] including during the perinatal period.[4] Previously published papers in this cohort have discussed the ethnic differences in detection of CMD within primary care, with some evidence that women of Pakistani ethnicity may be less likely to seek help for mental ill health.[20 40] Type 2 diabetes diagnoses in primary care records may also merely reflect the likelihood of screening; a recent systematic review, although

**Table 2** Cox regression of type 2 diabetes in women with versus without an indicator of antenatal CMD (N=909 women)

| | Type 2 diabetes | |
|---|---|---|
| | HR (95% CI) | P value |
| Unadjusted | | |
| Antenatal CMD Reference category=no indicator | | |
| Antenatal CMD indicator | 0.95 (0.57 to 1.57) | 0.839 |
| Adjusted* | | |
| Antenatal CMD Reference category=no indicator | | |
| Antenatal CMD indicator | 0.94 (0.56 to 1.58) | 0.823 |

Models using imputed data.
*Adjusted for maternal age, education, ethnicity, smoking and preconception CMD.
CMD, common mental disorders.

only using studies of US women, found that Asian women were more likely to be tested for type 2 diabetes following GDM than White women; pooled proportion being screened was 50% in Asian and 35% in White.[41] However, if an indicator of CMD or type 2 diabetes in primary care records merely did reflect a tendency to engage with health services, there would have been more evidence of an association between the two measures. Nonetheless, while in the UK it is recommended that women who have experienced GDM undergo glucose monitoring at 6 weeks postpartum and annually thereafter,[16] evidence suggests patchy implementation of this guidance.[42] Therefore there is possible underdiagnosis of type 2 diabetes following GDM among women in the UK.

While the variables of fasting glucose at the time of the OGTT and insulin use were investigated as potential moderators, there were no other measures of factors also known to be affected by the presence of CMD and which may impact on type 2 diabetes risk, such as adherence

with medication and lifestyle interventions. However, while most of the evidence of the adverse influence of depression on these behaviours comes from the type 2 diabetes population,[14] as discussed in the introduction, there remains very little evidence in women with GDM that mental health affects these behaviours. For example, a study of 400 women with GDM in Iran found no correlation between symptoms of anxiety and depression and 'self-care behaviours' such as dietary monitoring, physical activity and blood glucose monitoring.[43] Moreover, dietary and physical activity interventions trialled in obese pregnant women (although prior to the development of GDM) have not found antenatal depression to be a significant barrier to adherence.[44 45] Similarly, there are factors occurring in the postnatal period which may be associated with antenatal CMD, which may mediate the risk of type 2 diabetes and which were not explored. These include events in subsequent pregnancies such as future episodes of GDM and/or CMD, breast feeding,[46 47] postnatal CMD[48–50] and changes made to diet and levels of physical activity in the postpartum which in turn may influence weight.[8]

While not the focus of this study, future research could focus on investigating the relationship between maternal mental health and these other factors known to influence metabolic risk beyond the antenatal period. Future research may also examine the impact of treatment of CMD on type 2 diabetes risk in women with GDM, in light of some evidence that treatment of depression in those with type 2 diabetes may improve their glycaemic control.[51 52] While it could be argued that an indicator of CMD in primary care records makes it likely that the CMD was treated, such differences in outcomes between treated and untreated CMD could not be explored with the measures available in this sample. Nonetheless, the findings from this analysis that women with antenatal CMD are not at greater risk of type 2 diabetes following GDM than women without antenatal CMD could provide reassurance to these women, particularly if replicated in other samples.

### Author affiliations
[1]Section of Women's Mental Health, King's College London, London, UK
[2]South London and Maudsley NHS Foundation Trust, London, UK
[3]Born in Bradford, Bradford Teaching Hospitals NHS Foundation Trust, Bradford, UK
[4]Centre for Cardiovascular Science, University of Edinburgh, Edinburgh, UK
[5]Department of Child & Adolescent Psychiatry, King's College London, London, UK
[6]Department of Psychological Medicine, Kings College London, London, UK

**Acknowledgements** Born in Bradford is only possible because of the enthusiasm and commitment of all the children and parents in Born in Bradford. We thank all the participants, health professionals and researchers who have made Born in Bradford happen. We gratefully acknowledge the contribution of TPP and the TPP ResearchOne team in completing study participant matching to GP primary care records and in providing ongoing informatics support. This project is supported by the UK's National Institute for Health Research (NIHR) Applied Research Collaboration (ARC) for South London.

**Contributors** CAW conceived the research question, designed the study analysis plan analysed the data and wrote the first draft of the manuscript. GS extracted

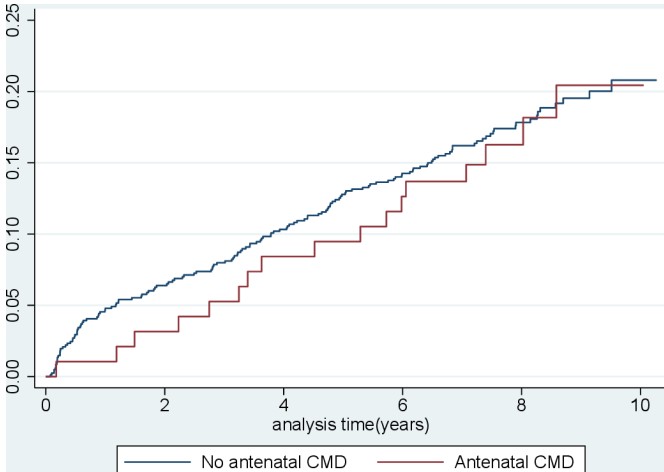

**Figure 2** Kaplan-Meier failure curves for time to development of type 2 diabetes (years) between women with (red line) and without (blue line) antenatal CMD. CMD, common mental disorders.

the data from primary care records, approved the data analysis plan and critically revised the draft. RR, ES, LH and KI approved the data analysis plan and critically revised the draft. All authors (CAW, GS, RR, ES, LH and KI) have approved the final version and agree to be accountable for all aspects of the work. CAW is the manuscript's guarantor.

**Funding** Born in Bradford receives core infrastructure funding from the Wellcome Trust (WT101597M) and a joint grant from the UK Medical Research Council (MRC) and Economic and Social Science Research Council (ESRC) (MR/N024397/1). CAW carried out this work as part of a UK Medical Research Council (MRC) funded Clinical Research Training Fellowship (MR/P019293/1). RR acknowledges the support of Tommy's and the British Heart Foundation (RE/18/5/34216). KI and LH receive salary support from the National Institute for Health Research (NIHR) Biomedical Research Centre at South London and Maudsley NHS Foundation Trust and King's College London. LH also receives salary support from the NIHR South London Applied Research Collaboration (ARC).

**Competing interests** None declared.

**Patient consent for publication** Not applicable.

**Ethics approval** Ethical approval for data collection in BiB was granted by Bradford Research Ethics Committee (Ref 07/H1302/112). Participants gave informed consent to participate in the study before taking part.

**Provenance and peer review** Not commissioned; externally peer reviewed.

**Data availability statement** Data are available on reasonable request. Information about access to data and collaborations available at https://borninbradford.nhs.uk/research/how-to-access-data/.

**ORCID iD**
Claire A Wilson http://orcid.org/0000-0003-2169-5115

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
