## [Reviewer comments · BMJ Open]

ARTICLE DETAILS

TITLE (PROVISIONAL)	Development of Type 2 diabetes in women with co-morbid gestational diabetes and common mental disorders in the Born in Bradford cohort
AUTHORS	Wilson, Claire; Santorelli, Gillian; Reynolds, Rebecca; Simonoff, Emily; Howard, Louise; Ismail, Khalida

VERSION 1 – REVIEW

REVIEWER	Chan, S-Y National University of Singapore, Department of Obstetrics and Gynaecology
REVIEW RETURNED	06-Aug-2021

GENERAL COMMENTS	This study has asked a new research question of whether the presence of the co-morbid condition of anxiety and depression during pregnancy could modulate the time taken to progress from gestational diabetes, diagnosed for the first time in the index pregnancy, to the later diagnosis of type 2 diabetes over the course of about 10 years. They have clearly presented the answer that it does not, with adjustment for appropriate covariates, and the main limitations of the study have been addressed in the discussion. The manuscript could be further improved by providing a bit more information. It is not entirely clear from reading the introduction why the authors hypothesised that a mood disorder that occurs only during the pregnancy, without necessarily persisting post-delivery, would be able to shorten the time to development of T2D. Is it because poor mood is expected to lead to poorer self-monitoring and control of glycaemia during pregnancy, a time of increased physiological insulin resistance, which can lead to irreversible or accelerated damage to organs that may participate in the development of increasing insulin resistance or pancreatic exhaustion over the longer term? The hypothesis in line 26-27 directs the reader to think that the absolute long term risk of T2D is postulated to increase in women with comorbid CMD rather than an increase in the time to development of T2D which is more the focus of this paper. In the results section and in table 2, it may help the reader comprehend the findings better if the point estimate of the time (95%CI) taken for 10% or 15% of each group to develop T2D is also presented alongside the hazard ratios. Discussion on the women's engagement with primary care providers post-delivery in the discussion section could be expanded more. Is there any information on the nationally
---

	recommended frequency of post-natal screening for T2D after a GDM pregnancy? If a recall system was operational to prompt women with a history of GDM to attend screening post-delivery, this should also be discussed. Infrequent screening could lead to delay in diagnosing T2D, under diagnosis of T2D and misclassification of undiagnosed T2D. Also having CMD could itself affect frequency of GP attendance in both directions – lack of attendance or increased attendance. Moreover evidence of GP engagement for follow-up for other medical conditions may not translate into adequate follow-up for a history of previous GDM. Is the assumption that the absence of a T2D code could signify both a negative screen and no attendance for screening? Can a distinction be made between attendance for a glucose evaluation which turned out to be negative as opposed to no screening for dysglycaemia, for example, by cross checking to see if blood tests for HbA1C or glucose was conducted? What was the criteria used to diagnosed T2D? Can it be assumed that T2D is diagnosed by the same criteria across primary care providers, and that the diagnosis cannot be in doubt? What is the distinction in the definition of “no” and “missing” preconception CMD? Is the absence of a record of CMD be considered “no” or “missing”? Similarly does “no” for Antenatal CMD mean an absence of a record of CMD? Should mention somewhere if the incidence of CMD in this cohort is comparable to other birth cohorts. Why only choose the specific mental health disorders of anxiety and depression to study and not the others? Disorders with psychotic elements may also pose challenges to the ability to self-organize lifestyle changes required to manage GDM well and to sustain improved lifestyle habits post-delivery. Also, were those with other mental health disorders (not anxiety and depression) included in the control group or excluded from the study? Was universal screening for mood disorders a practice occurring at the time of the study? Some of the drugs on the prescription list may also be used for non-mental health indications for example amitriptyline for chronic pain or insomnia. Were the prescriptions lined up against the diagnostic codes to ascertain if they were given for anxiety and depression? All of these could lead to misclassification errors of CMD which should be discussed. How common is it for women with anxiety and depression in pregnancy, particularly if mild, to not seek any medical help or only seek help from antenatal healthcare providers outside of the primary care setting and therefore not captured in GP records? IS the intention of the authors to just consider moderate to severe CMD? Data was imputed for 17.5% (n=159) cases. The amount of imputation undertaken should be made clear in the main manuscript, in addition to comments on sensitivity analyses showing similar results when only analyzing those with a full dataset. Which factors constituted the majority of the imputed variables? Could BMI confound the relationship being examined? Was BMI related to CMD or GDM and also to T2D? Table 1 should also include the antenatal fasting glucose, 2h glucose results and need for metformin/insulin treatment as a reflection of the severity of GDM or quality of glycaemic control.
--	---

	Minor points: Define CMD in abstract Line 16-19: what is the direction of effect of fasting glucose and glycaemic control on future T2D? Line 21: GDM is a highly heterogenous condition. Underlying mechanisms include insulin resistance and relative pancreatic insufficiency? Were cases confined to singletons or were there also multiple pregnancies (which are associated with greater insulin resistance)? Were all women in the BIB cohort universally screened for GDM? How exactly was date of conception calculated from LMP and dating scans? May be more appropriate to call it estimated date of conception since this date is never a certainty. Line 179-180 should be rephrased.
--	---

VERSION 1 – AUTHOR RESPONSE

This study has asked a new research question of whether the presence of the co-morbid condition of anxiety and depression during pregnancy could modulate the time taken to progress from gestational diabetes, diagnosed for the first time in the index pregnancy, to the later diagnosis of type 2 diabetes over the course of about 10 years. They have clearly presented the answer that it does not, with adjustment for appropriate covariates, and the main limitations of the study have been addressed in the discussion. The manuscript could be further improved by providing a bit more information.
Response: we thank you for taking the time to review our manuscript.

It is not entirely clear from reading the introduction why the authors hypothesised that a mood disorder that occurs only during the pregnancy, without necessarily persisting post-delivery, would be able to shorten the time to development of T2D. Is it because poor mood is expected to lead to poorer self-monitoring and control of glycaemia during pregnancy, a time of increased physiological insulin resistance, which can lead to irreversible or accelerated damage to organs that may participate in the development of increasing insulin resistance or pancreatic exhaustion over the longer term?

Response: yes, that is correct. This has now been made more explicit in the introduction.

Page 3 lines 26-28 now read:

“Thus it may be hypothesised that in women with GDM, co-morbid CMD during pregnancy could increase the risk of Type 2 diabetes in the future due to poorer glycaemic control during pregnancy.”

The hypothesis in line 26-27 directs the reader to think that the absolute long term risk of T2D is postulated to increase in women with comorbid CMD rather than an increase in the time to development of T2D which is more the focus of this paper. In the results section and in table 2, it may help the reader comprehend the findings better if the point estimate of the time (95%CI) taken for 10% or 15% of each group to develop T2D is also presented alongside the hazard ratios.

Response: thank you for this suggestion. This information has now been added to the results section as you suggest.

Page 7 lines 124-126 now read:

“The median time taken for 15% of women to develop Type 2 diabetes in the group unexposed to antenatal CMD was 6.5 years (95% CI 4.9, 7.9) versus 7.4 years (95% CI 3.6, 7.9) in the exposed group.”

Discussion on the women's engagement with primary care providers post-delivery in the discussion section could be expanded more. Is there any information on the nationally recommended frequency of post-natal screening for T2D after a GDM pregnancy? If a recall system was operational to prompt women with a history of GDM to attend screening post-delivery, this should also be discussed. Infrequent screening could lead to delay in diagnosing T2D, under diagnosis of T2D and misclassification of undiagnosed T2D. Also having CMD could itself affect frequency of GP attendance in both directions – lack of attendance or increased attendance. Moreover evidence of GP engagement for follow-up for other medical conditions may not translate into adequate follow-up for a history of previous GDM.

Is the assumption that the absence of a T2D code could signify both a negative screen and no attendance for screening? Can a distinction be made between attendance for a glucose evaluation which turned out to be negative as opposed to no screening for dysglycaemia, for example, by cross checking to see if blood tests for HbA1C or glucose was conducted?

Response: data such HbA1c and follow up glucose monitoring are not currently available in the dataset but the current literature suggests that postnatal follow up of women following GDM in the UK is extremely patchy. This is a significant limitation which has now been further acknowledged in the discussion section. Thank you for highlighting it.

Page 8 lines 168-172 now read:

“Nonetheless, while in the UK it is recommended that women who have experienced GDM undergo glucose monitoring at six weeks postpartum and annually thereafter, 16 evidence suggests patchy implementation of this guidance.³⁹ Therefore there is possible underdiagnosis of Type 2 diabetes following GDM among women in the UK.”

However, we have also stated in the discussion that the Type 2 diabetes diagnoses have reasonable face validity as the proportion of women developing Type 2 diabetes is similar to that found in other studies. The demographic predictors of Type 2 diabetes in this cohort also match the known risk factors e.g. ethnicity.

What was the criteria used to diagnosed T2D? Can it be assumed that T2D is diagnosed by the same criteria across primary care providers, and that the diagnosis cannot be in doubt?

Response: Type 2 diabetes is diagnosed in England according to the NICE (National Institute for Health and Care Excellence) guidelines by the presence of persistent hyperglycaemia and clinical symptoms. These guidelines have now been referenced in the methods.

What is the distinction in the definition of “no” and “missing” preconception CMD? Is the absence of a record of CMD be considered “no” or “missing”? Similarly does “no” for Antenatal CMD mean an absence of a record of CMD? Should mention somewhere if the incidence of CMD in this cohort is comparable to other birth cohorts.

Response: ‘no’ for antenatal CMD means that there were no prescription or Read codes for CMD in the mother's records between the estimated date of conception and date of delivery. ‘No’ for preconception means that there were no prescription or Read codes for CMD in the mother's records prior to the date of their first pregnancy (which may or may not have been a BiB pregnancy). If this date was unknown, preconception CMD was coded as ‘missing’. Further details of how the preconception codes were dated has now been added to supplementary information S1.

Why only choose the specific mental health disorders of anxiety and depression to study and not the others? Disorders with psychotic elements may also pose challenges to the ability to self-organize lifestyle changes required to manage GDM well and to sustain improved lifestyle habits post-delivery. Also, were those with other mental health disorders (not anxiety and depression) included in the control group or excluded from the study? Was universal screening for mood disorders a practice occurring at the time of the study? Some of the drugs on the prescription list may also be used for

non-mental health indications for example amitriptyline for chronic pain or insomnia. Were the prescriptions lined up against the diagnostic codes to ascertain if they were given for anxiety and depression? All of these could lead to misclassification errors of CMD which should be discussed. How common is it for women with anxiety and depression in pregnancy, particularly if mild, to not seek any medical help or only seek help from antenatal healthcare providers outside of the primary care setting and therefore not captured in GP records? IS the intention of the authors to just consider moderate to severe CMD?

Response: it is correct that there is likely underdiagnosis of common mental disorders in primary care, including during pregnancy. The proportion of women in the sample with an indicator of antenatal CMD is similar to that of other UK samples. The comment about face validity in the discussion has been expanded to make this more clear, while also acknowledging that this is likely an underestimate (an inherent limitation of primary care diagnoses). The ethnic differences in this population have also been discussed.

Page 8 lines 159-160 now read:

“a number of studies have documented the underdiagnosis of CMD in primary care,^{38 39} including during the perinatal period.⁴⁰”

NICE (National Institute for Health and Care Excellence) guidelines CG192

(<https://www.nice.org.uk/guidance/cg192>) on antenatal and postnatal mental health currently recommend that women are asked about symptoms of depression and anxiety in primary care and/or at booking with midwives.

As outlined in the methods, the codes used for CMD were from previously published methods, which have also been validated alongside a measure of mental health symptoms (Prady SL, Pickett KE, Petherick ES, et al. Evaluation of ethnic disparities in detection of depression and anxiety in primary care during the maternal period: Combined analysis of routine and cohort data. *British Journal of Psychiatry* 2016;208(5):453-61.) From this analysis it is not anticipated that misclassification is a significant source of bias in this study, particularly also given that the proportion of women in the sample with an indicator of antenatal CMD is similar to that of other UK samples.

Regarding other mental disorders e.g. psychotic disorders, also known as severe mental illness, they may present an additional risk to metabolic health as they are often treated with antipsychotic medication, which may increase the risk of diabetes (as opposed to antidepressants for which there is far less evidence of metabolic side effects). Thus it was hypothesised that the underlying mechanism for a relationship between antenatal SMI in women with GDM and subsequent Type 2 diabetes may be quite different to that from CMD.

However, given the substantial comorbidity between SMI and CMD, we have now removed 57 women with co-morbid SMI from the sample in a sensitivity analysis, which changed the results very little. Details of how SMI was coded have been added to the supplementary material (S2). The following text has also been added to the methods and results sections:

Page 5 lines 98-102 now read:

“A sensitivity analysis removing women with co-morbid severe mental illnesses (SMI) was also conducted. These relate to ICD-10 diagnostic groups F20-29 psychotic disorders including schizophrenia, F30-31 bipolar disorder and F60 personality disorders²¹ using GP electronic health records (EHR) alongside medication prescriptions as per previously published methods.²⁰ Further information on the codes used is provided in supplementary material (S2).”

Page 7 lines 133-135 now read:

“Removal from the sample of 57 women with co-morbid SMI resulted in little change to the effect estimate (unadjusted HR 0.95 (95% CI 0.55, 1.65) and adjusted HR 0.89 (95% CI 0.51, 1.56).”

Data was imputed for 17.5% (n=159) cases. The amount of imputation undertaken should be made clear in the main manuscript, in addition to comments on sensitivity analyses showing similar results when only analyzing those with a full dataset. Which factors constituted the majority of the imputed variables?

Response: page 5 line 95 states that:

“20 imputations were used according to the proportion of participants with any missing data”
(Bodner TE. What Improves with Increased Missing Data Imputations? Structural Equation Modeling: A Multidisciplinary Journal 2008;15(4):651-75)

Table 1 shows the proportion of missingness across each variable.

Complete case analyses are displayed in supplementary information S3 and page 7 line 131 states that:

“Results of complete case analyses mirrored those of the imputed results.”

Could BMI confound the relationship being examined? Was BMI related to CMD or GDM and also to T2D?

Response: confounders were considered carefully and rationalised (particularly considering the sample size and amount of missing data) according to the principles of confounder selection outlined by VanderWeele (VanderWeele TJ. Principles of confounder selection. European Journal of Epidemiology 2019;34(3):211-19) i.e. only those variables likely to be associated with both exposure (CMD) and outcome (Type 2 diabetes) were used. As a previous analysis in this sample had provided no evidence for an association between pre-pregnancy BMI and antenatal CMD, the decision was made not to include pre-pregnancy BMI as a confounder in this analysis.

Table 1 should also include the antenatal fasting glucose, 2h glucose results and need for metformin/insulin treatment as a reflection of the severity of GDM or quality of glycaemic control.

Response: fasting glucose and insulin use have been added to Table 1 as suggested as these variables were used in the analysis.

Minor points:

Define CMD in abstract

Response: the abbreviation CMD is not currently used in the abstract as journal guidance is to not use abbreviations in the abstract.

Line 16-19: what is the direction of effect of fasting glucose and glycaemic control on future T2D?

Response: inadequate glycaemic control during pregnancy following GDM diagnosis has been associated with increased future risk of Type 2 diabetes independently of baseline fasting glucose.

Line 21: GDM is a highly heterogenous condition. Underlying mechanisms include insulin resistance and relative pancreatic insufficiency?

Response: thank you for this suggestion. This now reads:

“There is also evidence of an increased risk in women whose GDM has required the use of insulin; this may be a marker of increased severity of insulin resistance, relative pancreatic insufficiency and/or poorer glycaemic control post diagnosis.”

Were cases confined to singletons or were there also multiple pregnancies (which are associated with greater insulin resistance)?

Response: 20 pregnancies were twin or triplet pregnancies (multiple pregnancies). Multiple pregnancy was not considered a significant confounder (see comments above about confounder selection) as there is limited evidence for an association between multiple pregnancy and increased risk of antenatal CMD (Howard LM, Molyneaux E, Dennis C-L, et al. Non-psychotic mental disorders in the perinatal period. The Lancet 2014;384(9956):1775-88.)

Were all women in the BIB cohort universally screened for GDM?

Response: yes.

Page 3-4 lines 46-47 read:

“It is offered to all women in Bradford, usually between 26 and 28 weeks gestation.”

How exactly was date of conception calculated from LMP and dating scans? May be more appropriate to call it estimated date of conception since this date is never a certainty.

Response: ‘date of conception’ has now been renamed ‘estimated date of conception’ as you suggest.

Line 179-180 should be rephrased.

Response: thank you for highlighting this. The word ‘however’ has been removed and this sentence now reads:

“These include events in subsequent pregnancies such as future episodes of GDM and/or CMD, breastfeeding, 43 44 postnatal CMD 45-47 and changes made to diet and levels of physical activity in the postpartum which in turn may influence weight.8”

VERSION 2 – REVIEW

REVIEWER	Chan, S-Y National University of Singapore, Department of Obstetrics and Gynaecology
REVIEW RETURNED	19-Jan-2022

GENERAL COMMENTS	All my points have been addressed adequately. The abbreviations of "CMD" and "GDM" in lines 25 and 26 of the abstract should be spelt out in full.
---